# Determination of oligomeric states of proteins via dual-color colocalization with single molecule localization microscopy

**Hua Leonhard Tan**[1], **Stefanie Bungert-Plümke**[1], **Daniel Kortzak**[1], **Christoph Fahlke**[1], **Gabriel Stölting**[1,2]*

[1]Institute of Biological Information Processing, Molecular and Cellular Physiology (IBI-1), Forschungszentrum Jülich, Jülich, Germany; [2]Berlin Institute of Health at Charité – Universitätsmedizin Berlin, Center of Functional Genomics, Hypertension and Molecular Biology of Endocrine Tumors, Charitéplatz, Berlin, Germany

**Abstract** The oligomeric state of plasma membrane proteins is the result of the interactions between individual subunits and an important determinant of their function. Most approaches used to address this question rely on extracting these complexes from their native environment, which may disrupt weaker interactions. Therefore, microscopy techniques have been increasingly used in recent years to determine oligomeric states in situ. Classical light microscopy suffers from insufficient resolution, but super-resolution methods such as single molecule localization microscopy (SMLM) can circumvent this problem. When using SMLM to determine oligomeric states of proteins, subunits are labeled with fluorescent proteins that only emit light following activation or conversion at different wavelengths. Typically, individual molecules are counted based on a binomial distribution analysis of emission events detected within the same diffraction-limited volume. This strategy requires low background noise, a high recall rate for the fluorescent tag and intensive post-imaging data processing. To overcome these limitations, we developed a new method based on SMLM to determine the oligomeric state of plasma membrane proteins. Our dual-color colocalization (DCC) approach allows for accurate in situ counting even with low efficiencies of fluorescent protein detection. In addition, it is robust in the presence of background signals and does not require temporal clustering of localizations from individual proteins within the same diffraction-limited volume, which greatly simplifies data acquisition and processing. We used DCC-SMLM to resolve the controversy surrounding the oligomeric state of two SLC26 multifunctional anion exchangers and to determine the oligomeric state of four members of the SLC17 family of organic anion transporters.

**\*For correspondence:**
gabriel.stoelting@bih-charite.de

**Competing interest:** The authors declare that no competing interests exist.

## Editor's evaluation

The authors present a method for measuring the average oligomerization state of fluorescently tagged membrane proteins by single-molecule localization microscopy (SMLM). In contrast to many other SMLM methods which aim to count subunits in membrane protein complexes, the authors aim to deduce the average oligomerization state from the probabilistic co-detection of at least 1 'reporter' fluorophore, which has relatively poor detection efficiency, with the detection of at least 1 fused 'marker' fluorophore. They calibrate the method against a set of proteins with known oligomerization states (validated against high-resolution clear native gel electrophoresis) and then apply it to convincingly clarify the oligomerization state of SLC26 and SLC17 family member membrane proteins. Although the approach is limited to measurements of the average oligomerization state, and as such is not suitable to measure a distribution of (higher) oligomerization states, it is nonetheless potentially very useful for identifying oligomerization states of unknown proteins in native cells, and furthermore works well with fluorophores that have poor detection efficiencies. The provided

software should be sufficient to allow other researchers with some experience in Python to perform this analysis on their own data.

## Introduction

Most membrane proteins form oligomeric assemblies, in which individual subunits cooperate to fulfill specialized functions. Some membrane proteins also form larger clusters, in which distinct classes of proteins interact with each other. The number of subunits within such protein structures is a crucial determinant of their function (*Lussier et al., 2019*). Native gel electrophoresis has been successfully used to determine oligomeric states of proteins and the molecular determinants of oligomeric assemblies (*Detro-Dassen et al., 2008*; *Gendreau et al., 2004*; *Schägger and von Jagow, 1991*). However, this technique relies on isolating protein complexes from their native environment, which can disrupt some protein interactions. To overcome this limitation, in situ methods such as fluorescence correlation spectroscopy (*Chen et al., 2003*), stepwise photobleaching (*Ulbrich and Isacoff, 2007*), and single molecule localization microscopy (SMLM) (*Annibale et al., 2011*; *Sengupta et al., 2011*) have become increasingly popular in recent years.

Stepwise photobleaching relies on the irreversible and stochastic photobleaching of fluorescent tags linked to the subunit to be counted, and the resulting intensity steps depend on the number of subunits within the complex being tested. Such data can be acquired using comparatively inexpensive microscopes which has undoubtedly led to the widespread adoption of this method (*Bartoi et al., 2014*; *Coste et al., 2012*; *Lussier et al., 2019*; *Ulbrich and Isacoff, 2007*). However, this approach is only feasible for low expression densities as it is limited by the resolution of light microscopy. Therefore, *Xenopus laevis* oocytes, which are large in diameter and in which ectopic protein expression can be controlled by adjusting the amount of injected RNA, are most often used for this method (*Ulbrich and Isacoff, 2007*). However, some proteins exhibit a different oligomeric state when expressed in the *Xenopus* system compared with mammalian expression systems (*Krashia et al., 2010*). For mammalian cells, the resolution limit of light microscopy can be circumvented using super-resolution microscopy methods (*Durisic et al., 2014*; *Hummer et al., 2016*; *Lee et al., 2012*; *Nan et al., 2013*; *Nicovich et al., 2017*; *Puchner et al., 2013*; *Sengupta et al., 2011*). SMLM increases resolution by reconstructing the precise positions of isolated fluorophore emissions detected over thousands of frames (*Betzig et al., 2006*; *Rust et al., 2006*). The fluorophores used for quantitative SMLM are usually photoactivatable or photoconvertible fluorescent proteins that are activated or converted at wavelengths different from that of the excitation laser (*Li and Vaughan, 2018*). Their positions in each frame may then be assigned to individual proteins in post-processing, to produce accurate information about the number and position of proteins subunits (*Sengupta et al., 2011*).

Several problems have to be overcome during image acquisition and processing to determine the oligomeric state of proteins: To increase the resolution in SMLM, the simultaneous activation of multiple fluorophores within the same diffraction-limited volume must be avoided. To assign signals to individual subunits within the same protein complex, the activation and emission of each fluorophore need to be well separated in time. This criterion can only be fulfilled if the probability of activating a certain protein is kept minimal through a gradually increased activation laser intensity. Algorithms to optimize laser control have been developed, but they are not intended to temporally separate clusters of emissions from individual proteins (*Lee et al., 2012*). During the long recordings required for SMLM, inherent mechanical and thermal fluctuations, as well as the sample drift, combine to reduce spatial resolution. The complex photophysics of fluorescent protein blinking may additionally result in over- or undercounting (*Annibale et al., 2011*; *Lee et al., 2012*). Another source of miscounting is the background noise, which cannot be easily distinguished from genuine signals at the single molecule level (*Shivanandan et al., 2014*; *Zhao et al., 2018*). Furthermore, it is impossible to detect all fluorophores in the sample because the probability of having a fluorescent marker detected during the imaging period (i.e., the recall rate) is always less than one. This likely arises from a combination of incomplete protein translation or misfolding, failure to get activated or converted, and pre-imaging bleaching (*Durisic et al., 2014*; *Nicovich et al., 2017*). Thus, the observed counts are not equal to the oligomeric state of interest but instead follow a binomial distribution that depends on both the real number of subunits per complex and the recall rate. This binomial strategy demands a recall rate of about 0.5or higher to ensure reasonable resolution between different oligomeric states (*Durisic et al.,*

*2014*; *Ulbrich and Isacoff, 2007*). Such a demand for high recall rates poses challenges with respect to fluorescent protein selection, sample preparation, data acquisition, and post-imaging data processing (*Shivanandan et al., 2014*). These criteria and drawbacks have limited the application of SMLM on counting.

Here, we developed a new strategy, dual-color colocalization SMLM (DCC-SMLM), which utilizes two spectrally different fluorescent proteins attached to the same protein of interest (POI). DCC-SMLM is very tolerant of the background noise and low recall rates of fluorescent proteins and does not require stringent control of the activation laser. We tested the new DCC-SMLM approach on proteins with known subunit composition, and then used it to study proteins from two plasma membrane transporter families with controversial or unknown oligomeric states: the SLC26 and SLC17 families.

The human SLC26 family has 10 members that function as anion transporters or channels as well as the motor protein in cochlear outer hair cells. The physiological importance of these proteins is illustrated by the growth defects, chronic diarrhea or deafness associated with naturally occurring mutations. Biochemical and recent structural studies indicated that the proteins exist as dimers (*Detro-Dassen et al., 2008*; *Geertsma et al., 2015*; *Walter et al., 2019*), whereas other studies have proposed a tetrameric state (*Hallworth and Nichols, 2012*; *Sinha et al., 2010*; *Zheng et al., 2006*). The SLC17 family includes proton-driven vesicular glutamate, aspartate, sialic acid, and nucleotide transporters, as well as $Na^+$-coupled phosphate transporters. Using DCC-SMLM, we determined the oligomeric states for two SLC26 proteins, the human epithelial chloride-bicarbonate exchanger SLC26A3 and the rat motor protein prestin (Slc26a5), and for four members of the SLC17 family: mouse vGlut1, -2, and -3 (Slc17a7, Slc17a6, and Slc17a8, respectively) and human sialin (SLC17A5).

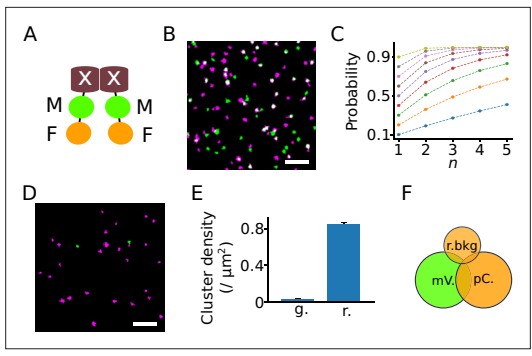

**Figure 1.** The dual-color colocalization (DCC) strategy to determine the oligomeric state of proteins. (**A**) Model of a dimeric fusion protein used for DCC-single molecule localization microscopy (SMLM). The protein of interest is marked with a white 'X', whereas the marker and indicator fluorescent proteins are labeled 'M' and 'F', respectively. (**B**) A reconstructed SMLM image from an HEK293T cell expressing $K_{ir}2.1$ labeled with both mVenus (green) and PAmCherry (magenta). White indicates colocalization of the two color signals. Scale bar: 1 μm. (**C**) The probability of detecting a protein complex changes with the number (*n*) and recall rate (*p*) of the fluorescent proteins it contains. Each color indicates a different recall rate from 0.1 to 0.9 in steps of 0.1. (**D**) A representative image reconstructed from a recording of a bare coverslip surface showing background signals in the PAmCherry (magenta) and mVenus channels (green). Scale bar: 1 μm. (**E**) The mean background cluster densities (± SEM) in the mVenus (green, g.) and the PAmCherry (red, r.) recording channels. A cluster comprises a series of fluorescent signals that are spatially close and clustered with the DBSCAN (density-based spatial clustering of applications with noise) algorithm. *N*=283 recordings. (**F**) As the background signals in the green channel are negligible, only three types of signals are considered: signals from mVenus (mV.), signals from PAmCherry (pC.), and background signals in the red channel (r.bkg).

The online version of this article includes the following figure supplement(s) for figure 1:

**Figure supplement 1.** Sensitivity analysis of *Equation 1* and *Equation 4* for different oligomeric states as indicated in the titles.

**Figure supplement 2.** Fraction of clusters remaining after the removal of large clusters (% after filter) by our analysis pipeline for each protein of interest in our study.

## Results
## DCC-SMLM for determining oligomeric states

A protein complex of *n* identical subunits, each labeled with a single indicator fluorescent protein (F) with a recall rate of *p*, will be detected with a probability $P_d$ as

$$P_d = 1 - (1 - p)^n \qquad (1)$$

In theory, $P_d$ is equal to the ratio of the number of proteins with visible fluorescence ($N_F$) to the number of labeled proteins in total ($N$), namely,

$$P_d = \frac{N_F}{N} \tag{2}$$

However, direct determination of $P_d$ from single-color microscopy images is impossible, as the total number of proteins, including those not exhibiting fluorescence, is unknown. Therefore, in order to experimentally determine $P_d$, we additionally labeled each protein subunit with another fluorescent protein (M) as a baseline marker (**Figure 1A**). As for F, the number of proteins observable by the fluorescence of M ($N_M$) will be smaller than $N$. Assuming that the fluorescence of F and M is independent, $P_d$ can be determined from the ratio of the number of proteins showing fluorescence from both M and F ($N_{MF}$) to that of the proteins showing fluorescence from M ($N_M$),

$$P_d = \frac{N_{MF}}{N_M} \tag{3}$$

Both $N_{MF}$ and $N_M$ can be counted from dual-color microscopy images (**Figure 1B**). The values of $P_d$ obtained for proteins with known oligomeric states were used to solve **Equation 1** for $p$, which could then be used to determine unknown oligomeric states (**Figure 1C**).

Independence of both fluorescence signals was guaranteed by using proteins with separate emission and excitation spectra as well as performing sequential rather than simultaneous imaging of color channels. However, based on our experience with solubilizing ion channels and transporters tagged with fluorescent proteins (**Guzman et al., 2022**; **Stölting et al., 2014**; **Stölting et al., 2015a**; **Tan et al., 2017**), we expected that a fraction of fluorescent proteins might be cleaved off within the linker region or within M, leading to the simultaneous removal of both F and M. We introduced an additional factor, $m$, to describe the probability of this phenomenon occurring. Notwithstanding that we cannot exclude other possibilities contributing to the loss of both fluorescent proteins, $m$ remains generally as the probability of both M and F simultaneously being dysfunctional. Dysfunction of F alone is still represented by the factor $p$. With $m$ included, the probability $P_d$ was hence described as

$$P_d = \sum_{k=1}^{n} \left\{ \frac{\binom{n}{k}(1-m)^k m^{n-k}}{1-m^n} \left[ 1 - (1-p)^k \right] \right\} \tag{4}$$

To assess how the values of $m$ and $p$ might alter the performance of the DCC strategy, an in silico sensitivity analysis based on **Equation 4** demonstrated that the optimal separation from monomers to tetramers roughly occurred in the range $0.2 < p < 0.4$, while $m$ should best remain below 0.3 (**Figure 1—figure supplement 1**).

In SMLM, diffraction-limited video frames, which capture the temporally isolated emission events of individual fluorophores, are fit to yield coordinates associated with additional attributes such as intensity and precision of the fit. These multi-feature coordinates are termed 'localizations' and are used to reconstruct super-resolution images after processing (**Betzig et al., 2006**). The emission from individual fluorophores typically remains active over several frames, depending on the video acquisition rate and fluorophore kinetics (e.g., for **Durisic et al., 2014**; **Subach et al., 2009b**), resulting in a cluster of localizations per emission event. Ideally, when the sample drift and chromatic aberrations are well corrected, all emission events from a single point emitter (e.g., a fluorescent protein) in a fixed sample fall into the same cluster.

We chose mVenus (**Nagai et al., 2002**) as the fluorescent marker protein M because of the very low background signal in its detection channel (**Figure 1D–E**). mVenus is derived from yellow fluorescent protein with several improvements such as faster maturation, a monomeric structure, and high tolerance to cellular acidification, and the halide concentration (**Nagai et al., 2002**). Like other GFP derivatives, it blinks by cycling between fluorescent and dark states under continuous illumination with the excitation laser (**Dickson et al., 1997**). PAmCherry was chosen as the indicator because it is quickly bleached by continuous excitation after activation (**Durisic et al., 2014**). However, this channel had considerably more background signals.

To deal with background signals, we partially corrected for the background by setting a minimum number of localizations for clustering (see Materials and methods). However, some background signals were still indistinguishable from genuine PAmCherry signals, which explains the higher background level in that color channel (*Figure 1D–F*). To reduce the contribution of the background noise to the final result, as illustrated in *Figure 1F*, we corrected $P_d$ as

$$P_\mathrm{d} = \frac{N_\mathrm{MF} - N_\mathrm{M}P_\mathrm{rb}}{N_\mathrm{M} - N_\mathrm{M}P_\mathrm{rb}} \tag{5}$$

in which $P_\mathrm{rb}$ is the probability of the mVenus cluster colocalizing with red background clusters. For any given mVenus cluster, the number ($N_\mathrm{rb}$) of background clusters from the red channel located in its radial vicinity $d$ follows the Poisson distribution: $N_\mathrm{rb} \sim \mathrm{Pois}(\pi d^2 D_\mathrm{rb})$, in which $D_\mathrm{rb}$ is the average density of red background clusters measured from the area on the coverslip outside of the cell. Therefore, we have

$$P_\mathrm{rb} = 1 - e^{-\pi d^2 D_\mathrm{rb}} \tag{6}$$

To prevent miscounting due to protein aggregation, we excluded the small fraction of clusters with large diameters (>500 nm) (*Figure 1—figure supplement 2*).

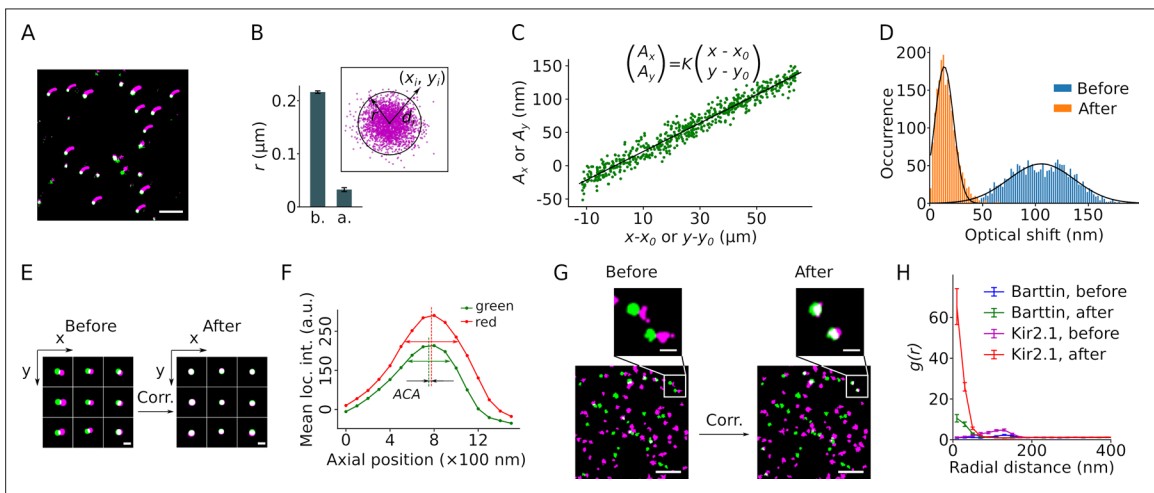

**Figure 2.** Correction of sample drift and chromatic aberration. (**A**) A single molecule localization microscopy (SMLM) image of a bead sample reconstructed from 13,000 frames recorded over ~18.5 min. The small magenta smears indicate the trajectory of the sample drift. The corrected positions are displayed in green or, when overlapping, in white. Scale bar: 1 μm. (**B**) The mean cluster radius of positions extracted from 41 beads before (b.) and after (a.) drift correction, with error bars indicating the SEM. The cluster radius is defined as $\mu + 2\sigma$ of the Euclidean distances ($d_i$) of all localizations (with coordinates ($x_i$, $y_i$)) to the center (mean coordinates) of the cluster, as shown in the inset. (**C**) Plot of the lateral chromatic aberration between the red and green channels, ($A_x$, $A_y$), versus the distance from the red coordinates to an arbitrary center point ($x_0$, $y_0$). Data from both dimensions are included in the graph. Linear regression (black line) yielded the values of the slope $K$, $x_0$, and $y_0$. The fit is a representative from a single recording of a bead sample. (**D**) Lateral chromatic aberrations from nine recordings of a bead sample with a total of 1980 beads were corrected with the values obtained in (**C**), with blue and orange indicating before and after the correction, respectively. Black lines show the Gaussian fitting. (**E**) Reconstructed images of nine representative beads from different areas before and after drift and chromatic aberration corrections. The green and magenta indicate signals from the mVenus and PAmCherry channels, respectively. Scale bar: 200 nm. (**F**) The mean intensities of the beads recorded at the green and the red channels changed along the axial position of the sample. The axial distance between the two peaks indicates the axial chromatic aberration (ACA). (**G**) Representative reconstructed images showing the mVenus (green) and PAmCherry clusters (magenta) recorded from HEK293T cells expressing barttin-mVenus-PAmCherry before and after drift and chromatic aberration corrections. Non-clustered localizations were omitted from the images. Scale bars indicate 1 μm and 200 nm respectively in the bottom and the zoomed-in images. (**H**) Radial distribution of barttin (17 cells, 1 preparation) and $K_{ir}2.1$ (35 cells, 3 preparations) PAmCherry clusters around mVenus clusters before and after drift and chromatic aberration corrections. Non-clustered localizations were excluded from the analysis.

The online version of this article includes the following figure supplement(s) for figure 2:

**Figure supplement 1.** Regional averaging for the correction of lateral chromatic abberation.

eLife Research article

## Correction of sample drift, chromatic aberrations, and definition of colocalization

The resolution of SMLM is compromised by sample drift and optical aberrations, which affect the determination of colocalization. We recorded sample drift using TetraSpeck beads as fiducial markers and corrected the drift during post-imaging data processing (*Yi et al., 2016*). This correction reduced the radius of clustered signals to ~30 nm (*Figure 2A and B*). Since our approach relies on colocalization of the marker and the indicator in two color channels, we also corrected chromatic aberrations resulting from different refractive indices at different emission light wavelengths. We confirmed earlier reports that lateral chromatic aberration (LCA) can be described as a set of linear functions centered on the optical axis of the microscope (*Erdelyi et al., 2013*; *Kozubek and Matula, 2000*). Values from a linear fit of the measured inter-color distances of individual beads to their coordinates were used for correction (*Figure 2C*). This way the LCA could be reduced to 13.8±8.9 nm for bead samples (*Figure 2D and E*). Chromatic aberrations may also be corrected by determining regional averages with similar results (*Figure 2—figure supplement 1*; *Churchman and Spudich, 2012*; *Georgieva et al., 2016*). The axial chromatic aberration (ACA) determined as the focus offset between colors was less than 100 nm and thus considered negligible for SMLM signal registration in only two dimensions (*Figure 2F*). Unlike fluorescent proteins , beads show an intense and long-lasting fluorescent signal. To evaluate the correction in fixed cell samples and to determine the offset threshold for colocalization, we applied the correction to cell samples expressing monomeric barttin (*Scholl et al., 2006*; *Estévez et al., 2001*), or tetrameric $K_{ir}2.1$ (*Kcnj2*, *Doyle et al., 1998*), each subunit tagged with a copy of mVenus and PAmCherry (*Figure 2G*). We calculated the radial distribution function (RDF) of the red clusters around each mVenus cluster (*Figure 2H*). Prior to the correction of chromatic aberrations, the RDFs from both proteins showed high densities peaked at a distance of about 150 nm. Following correction, highest densities of red clusters were found within a radius of ~100 nm around mVenus clusters for both proteins, representing the achievable two-color resolution of our experimental setup. We therefore chose 100 nm as the threshold distance for colocalization.

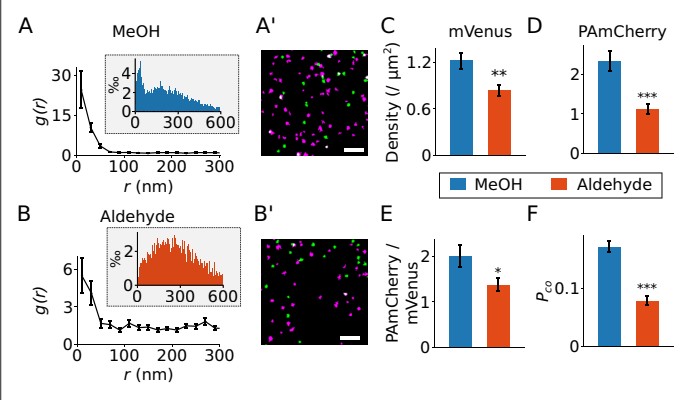

**Figure 3.** Effect of different fixation methods. (**A and B**) Results from the radial distribution function g(r) for PAmCherry clusters around mVenus clusters recorded in Flp-In T-REx 293 cells expressing barttin-mVenus-PAmCherry and fixed with cold methanol (A, MeOH, 24 cells) or para-formaldehyde plus glutaraldehyde (B, aldehyde, 29 cells). The inset shows the histograms of the nearest neighbor distances. (**A' and B'**) Reconstructed images from recordings of cells expressing barttin-mVenus-PAmCherry fixed with cold methanol (**A'**) or the aldehyde (**B'**), with green indicating mVenus clusters and magenta for PAmCherry clusters. Scale bar: 1 μm. (**C and D**) The densities of mVenus (C, p=0.0028) and PAmCherry (D, p=4.9 × 10⁻⁵) clusters in samples prepared using the two fixation methods. (**E**) The relative density of PAmCherry (pC.) clusters to mVenus (mV.) clusters, p=0.02. (**F**) The measured colocalization ratio of mVenus with PAmCherry, p=1.6 × 10⁻¹⁰. *: 0.0 5 > p > 0.01, **: 0.01 > p > 0.001, ***: p<0.001, Student's t-test. Data is shown as mean ± SEM.

The online version of this article includes the following figure supplement(s) for figure 3:

**Figure supplement 1.** The order of fluorescent proteins does not affect the colocalization ratio.

**Table 1.** Colocalization ratio ($P_d$) of mVenus clusters with PAmCherry clusters of fusion proteins fixed with PFA+GA or cold methanol.

| Protein | PFA +GA | | Cold methanol | |
|---|---|---|---|---|
| | $P_d$ | $N$ | $P_d$ | $N$ |
| ClC-2 | 0.215 | 19 | 0.237 | 32 |
| bClC-K | n/a | n/a | 0.241 | 15 |
| EAAT2 | n/a | n/a | 0.293 | 22 |
| Kcnj2 (K$_{ir}$2.1) | 0.248 | 23 | 0.411 | 35 |
| SLC26A3 | n/a | n/a | 0.214 | 26 |
| Prestin | 0.176 | 27 | 0.251 | 33 |
| vGlut1 | 0.119 | 25 | 0.170 | 23 |
| vGlut2 | 0.111 | 25 | 0.165 | 25 |
| vGlut3 | 0.131 | 26 | 0.163 | 32 |
| sialin | n/a | n/a | 0.172 | 22 |
| barttin | 0.078 | 27 | 0.172 | 17 |

## Fixation conditions and the fusion pattern of the fluorescent proteins

A critical prerequisite for DCC studies is to sufficiently immobilize the studied proteins while preserving the function of fluorescent proteins. A large proportion of membrane proteins remain mobile after fixation with paraformaldehyde (PFA) alone, requiring the addition of glutaraldehyde (GA) or the use of cold methanol (MeOH) for adequate immobilization (*Stanly et al., 2016*; *Tanaka et al., 2010*). To test whether these fixation methods affect mVenus and PAmCherry fluorescence in SMLM, we stably expressed the ClC-K β-subunit barttin (*Estévez et al., 2001*) C-terminally tagged with both PAmCherry and mVenus. Both fixation methods led to comparable correlation distances between the two colors (*Figure 3A and B*). However, upon MeOH fixation we observed a significantly higher signal density, a higher ratio of PAmCherry clusters to mVenus clusters and a higher colocalization ratio than with the aldehyde fixation (*Figure 3A', B' and C–F*). Similar results were obtained for many of the other proteins used in this study (*Table 1*). Therefore, we concluded that fixation with cold methanol better preserved the function of the fluorescent proteins compared with PFA/GA fixation and, therefore, we chose cold methanol as our routine fixation reagent.

Since proteins of interest (POIs) were tagged with both PAmCherry and mVenus, we investigated whether the fusion pattern affected the outcome. We compared the colocalization ratios when PAmCherry and mVenus were tagged to the C-terminus of POIs in two different ways: either PAmCherry or mVenus preceding the other. The results, as shown in *Figure 3—figure supplement 1*, indicated that the fusion pattern did affect the colocalization ratio.

## Calibration of the DCC model

Given that the recall rate (*p*) of fluorescent proteins may differ between experimental setups (*Durisic et al., 2014*; *Wang et al., 2014*), we first calibrated the DCC model (*Equations 4–6*) using standards with known oligomeric state: the monomeric ClC-K chloride channel β-subunit barttin (*Scholl et al., 2006*; *Estévez et al., 2001*), the dimeric chloride channels ClC-2 and ClC-K (*Park et al., 2017*; *Stölting et al., 2014*; *Thiemann et al., 1992*), the trimeric glutamate transporter EAAT2 (*Canul-Tec et al., 2017*; *Nothmann et al., 2011*) and the tetrameric potassium channel K$_{ir}$2.1 (Kcnj2, *Doyle et al., 1998*). All proteins were C-terminally tagged with PAmCherry and mVenus via two short peptide linkers (*Table 2*).

As an independent confirmation of the reported oligomeric state as well as of the function of our constructs, we performed high-resolution clear native electrophoresis (hrCNE) with these proteins

**Table 2.** The linkers used in the fusion of proteins of interest (POIs) to the fluorescent protein.

| Name | Amino acid sequence | Length (a.a.) |
|---|---|---|
| Linker #1 | GGSGG PSGLR SGSGG GSASG GSGS | 24 |
| Linker #2 | PPVGT ELGST | 10 |
| Linker #3 | GGSGG PGGSG GSPVG TELGS T | 21 |
| Linker #4 | GSGSG GGSAS GGSGS | 15 |

In this project two fusion patterns were tested: POI-Linker #1-PAmCherry-Linker #2-mVenus (r-g) and POI-Linker #3-mVenus-Linker #4-PAmCherry (g-r) (both written from the N- to C-terminus) with the following linker sequences.

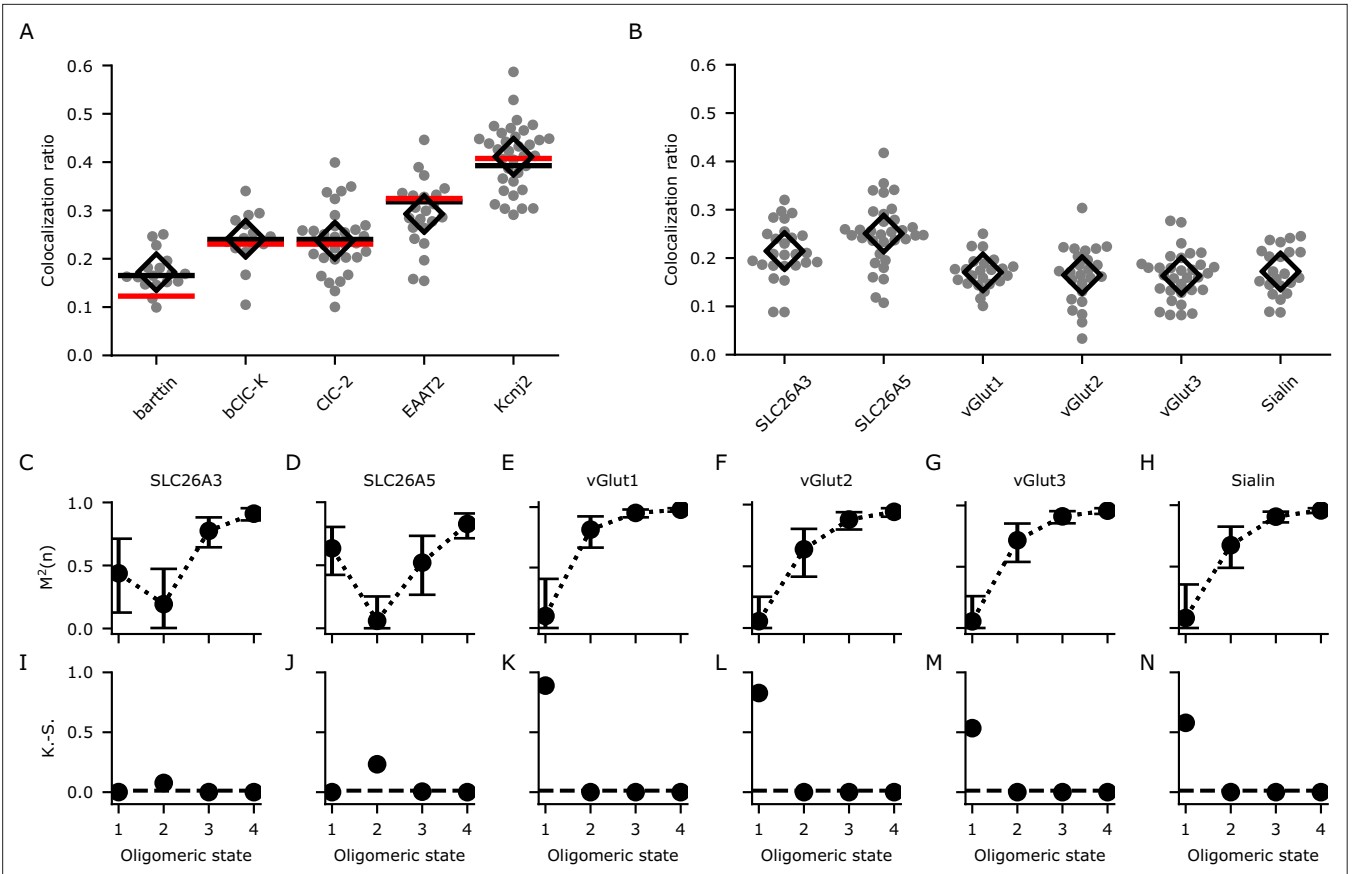

**Figure 4.** Calibration and determination of oligomeric states through analyzing the colocalization ratios with dual-color colocalization (DCC)-single molecule localization microscopy (SMLM). (**A**) The colocalization ratios for mVenus and PAmCherry linked to proteins with known oligomeric states for calibration. Red lines show the fit with *Equation 1* and black lines show the fit with *Equation 4* including the modification factor *m*. The number of cells included in the analysis are (in brackets): barttin (13), ClC-2 (30), bClC-K (14), EAAT2 (19), and $K_{ir}2.1$ (34). Individual values are shown as gray circles and the mean value is shown as a black open diamond. (**B**) Colocalization ratios for mVenus and PAmCherry linked to the indicated proteins of interest. The number of cells included are 24, 27, 22, 25, 30, and 22, respectively. (**C–H**) The coefficient of mismatch ($M^2$) calculated for all the proteins of interest at assumed oligomeric states of monomer (1), dimer (2), trimer (3), and tetramer (4). Error bars correspond to the range of the 95% confidence intervals as determined by a global bootstrap resampling (10,000 resamples). (**I–N**): Two-sample Kolmogorov-Smirnov (K.-S.) tests of the observed colocalization ratios from each protein of interest compared with the protein standards in (**A**). The dashed horizontal line indicates the Bonferroni-corrected threshold for the α-level (0.0125).

The online version of this article includes the following source data and figure supplement(s) for figure 4:

**Figure supplement 1.** Determination of the most likely oligomeric state of SLC26 and SLC17 proteins using only *Equation 1*.

**Figure supplement 2.** High-resolution clear native gel electrophoresis (hrCNE) of reference proteins and proteins of interest.

**Figure supplement 2—source data 1.** This source data contains the unedited gel images.

**Figure supplement 3.** The working paradigm of dual-color colocalization (DCC)-single molecule localization microscopy (SMLM).

**Figure supplement 4.** Summary of 1000 simulations per data point of a mixed population of dimeric and tetrameric proteins.

**Figure supplement 5.** Plots of the experimentally observed expression densities versus the colocalization ratio do not reveal a consistent correlation.

(*Figure 4—figure supplement 2*). Our results also demonstrate that the two linked fluorescent proteins did not interfere with the reported quaternary structure.

After correction for sample drift and chromatic aberrations, localizations were assigned to clusters using the DBSCAN (density-based spatial clustering of applications with noise) algorithm (*Ester et al., 1996*). Background subtracted colocalization ratios for mVenus and PAmCherry clusters were then calculated (*Equations 5 and 6*, *Figure 4A*). We used the known oligomeric states (*n*) to obtain the recall rate (*p*) of PAmCherry from *Equation 1*, or *p* and the modification factor (*m*) from *Equation 4* of the DCC model. A global fit of bootstrap resampled (10,000 resamples) experimental data

from all five proteins with known oligomeric states to *Equation 1* suggested a low recall rate (*p*) of 0.12 (95% confidence interval [CI]: 0.12–0.13) for PAmCherry. In general, the fit did describe the data reasonably well ($R^2$ = 0.95) but showed a large deviation for the monomeric reference protein as expected (*Figure 4A*). Fitting to *Equation 4* gave values of 0.30 (95% CI: 0.21–0.39) for *m* and 0.17 (95% CI: 0.15–0.18) for *p*. As expected, this model only aligns slightly better with the data ($R^2$ = 0.97) compared to the simpler model (*Equations 1–3*). However, the improvement is especially visible for the monomeric reference protein (*Figure 4A*), and we therefore chose *Equation 4* to analyze proteins with contradictory or unknown oligomeric states.

## Using DCC-SMLM to determine the oligomeric states of SLC26 and SLC17 proteins

We next used DCC-SMLM to resolve the oligomeric states of several SLC26 and SLC17 proteins. The colocalization ratios of mVenus and PAmCherry clusters for these proteins were determined (*Figure 4B*) and compared with those expected for different oligomeric states using the DCC model (*Equation 4*). To quantify the difference between expected and observed values, we introduced the coefficient of mismatch $M^2$ (*Equation 15*). An $M^2$ value of 0 indicates a perfect match, with larger values indicating larger differences. The calculated value is dependent on an accurate determination of *p* and *m* in *Equation 4* as well as of the colocalization ratios of the POIs. To include the combined error of these measurements, we used bootstrap resampling over all of these parameters to calculate means and 95% confidence intervals. As shown in *Figure 4C-H*, $M^2$ values reached their lowest values for a dimeric state of SLC26 proteins and for a monomeric state of all SLC17 proteins. As predicted for a monomer, accurate determination with *Equation 1* alone (only *p*, no *m*) was not feasible as seen for the SLC17 proteins (*Figure 4—figure supplement 1*).

Even using *Equation 4*, the confidence interval at the dimeric state for SLC26A3 overlapped with that of the monomeric state (*Figure 4C*), preventing the confident designation as a dimer based on the $M^2$ values alone. To resolve these uncertainties, we additionally used a two-sample Kolmogorov-Smirnov (K.-S.) test (*Figure 4I–N*), which allows for the determination of the most likely oligomeric state by comparing the colocalization ratios from the POI with those from proteins with known stoichiometry, thereby foregoing the need to determine *p* and *m*. With this test, we confirmed the results of the $M^2$ values and also demonstrated that SLC26A3 is most likely a dimeric protein as it showed a significantly different distribution compared to reference proteins with other oligomeric states (*Figure 4I*). Moreover, we independently confirmed our results for the oligomeric states of SLC26 and SLC17 proteins with hrCNE utilizing the same constructs and expression system as in our DCC-SMLM experiments (*Figure 4—figure supplement 2*).

## Discussion

We have developed a novel DCC-SMLM technique to determine the oligomeric state of plasma membrane protein complexes in situ in mammalian cells. Using mVenus as the marker fluorescent protein and PAmCherry as the indicator fluorescent protein for counting, we established a dimeric state for two representatives of the SLC26 family, the epithelial anion exchanger SLC26A3 and the motor protein prestin (Slc26a5); and a monomeric state for four representatives of the SLC17 family, the vesicular glutamate transporters vGlut1, -2, and -3 and the lysosomal sialic acid transporter sialin.

Our DCC-SMLM approach (*Figure 4—figure supplement 3*) overcomes several limitations of the methods commonly used to determine the oligomeric state of proteins. Unlike for biochemical methods, no solubilization or purification of proteins is necessary as DCC-SMLM works with proteins in situ. In comparison to photobleaching experiments, DCC-SMLM can be applied reliably even in small mammalian cells where the high densities of protein expression are usually problematic. In contrast to previously reported quantitative SMLM methods, DCC-SMLM simplifies the procedure and offers greater reliability of counting. Binomial SMLM counting strategies require clustering of signals in at least three dimensions (*x*, *y*, and time) to identify individual protein subunits. This demands stringent control of the activation laser and intensive post-imaging data processing (*Lee et al., 2012*). Another recently published SMLM-based quantification strategy also foregoes the need for temporal clustering as it utilizes the blinking behavior of some fluorescent proteins (*Fricke et al., 2015*; *Hummer et al., 2016*). This, however, requires fluorescent proteins that show blinking behavior (multiple on

**Table 3.** The settings used for single molecule localization microscopy (SMLM) imaging.

| Channel | FP | Excitation laser | Emission filter | Collection range |
|---|---|---|---|---|
| Green | mVenus | 514 nm @ 6.1/4.4 mW | FF01-485/537/627−25+FF03-525/50-25 | 526–546 nm |
| Red | PAmCherry | 561 nm @ 6.5/5.4 mW | FF01-609/57-25 | 580–537 nm |

The dichroic mirror Di01-R442/514/561−25x36 was used for imaging of both fluorescent proteins. All filters and dichroic mirrors were acquired from Semrock. The laser power was measured at the sample plane and before the rear entrance of the objective. The power of the 405 nm activation laser for PAmCherry was varied between 4.4 µW and 10.2 mW near the rear entrance, corresponding to 3.0 µW to 4.8 mW in the sample plane.

↔ off sequences) which is not guaranteed for all fluorescence proteins or dyes (*Durisic et al., 2014*; *Subach et al., 2009a*). Additionally, all previously published SMLM quantification and photobleaching step analysis methods require large recall rates ($p \geq 0.5$) as they are otherwise unable to resolve the small oligomers typically observed for most membrane proteins. In contrast, DCC-SMLM only relies on the relationship between the total detection probability for a protein complex and the number of fluorophores within the complex, as displayed in *Equations 1 and 4*. This relationship is independent of temporal separation or demanding fluorophore properties, such as high recall rates, thereby simplifying the data acquisition and post-processing.

At the single molecule level, noise and genuine signals become hard to distinguish, especially for photoactivatable fluorescent proteins with low brightness such as PAmCherry (*Subach et al., 2009b*). Since DCC considers the overlap of two colors, the effect of background signals on the indicator is greatly reduced. The probability of noise signals falling within the colocalization threshold distance of a given mVenus cluster follows a Poisson distribution. Our measured background signal density in the PAmCherry channel of ~0.8 clusters/µm² inside cells (*Figure 1E*) suggests that only 2.6% of all mVenus molecules erroneously colocalize with the background clusters from the red channel (*Equation 6*). Given the colocalization ratios greater than 0.2 (for dimers and higher oligomeric forms; *Figure 4*), this noise level results in a signal-to-noise ratio greater than 10. This estimate compares favorably with single-color experiments: transfected cells showed a mean density of ~2.2 PAmCherry clusters/µm² consistent with a signal-to-noise ratio less than 4 for this fluorescent protein alone. Higher background noise levels in the marker detection channel would lead to the over-counting of marker clusters, resulting in a falsified colocalization ratio and should be avoided. Our use of the bright mVenus tag as the marker allowed us to use a narrow detection window, resulting in very low levels of background noise (*Figure 1D* and *Table 3*).

For data processing, filtering of noise signals is important for determining the recall rate accurately. The higher tolerance to background noise enabled us to solely use a simple filter based on an empirical threshold of the minimal number of localizations for a single PAmCherry cluster (*Figure 5*). This was based on our observation that signal clusters recorded from background regions showed mostly only small clusters with few localizations (*Figure 5*). Fusion of two fluorescent proteins linked sequentially did not interfere with their structure as indicated by our hrCNE results (*Figure 4—figure supplement 2*). This may not be surprising as similar results have been reported previously for similar constructs in other contexts (*Leeman et al., 2018*; *Liang et al., 2018*; *Murata et al., 2005*; *Ranade et al., 2014*).

In conventional binomial analyses of SMLM data, a low recall rate ($p<0.5$) is a significant limitation when determining oligomeric states of proteins (*Durisic et al., 2014*). Theoretically, DCC-SMLM can optimally distinguish oligomeric states from monomer to pentamer if $p$ is between 0.2 and 0.4 (*Figure 1C*; *Figure 1—figure supplement 1*), dependent on the oligomeric state of interest. This makes PAmCherry and other fluorescent proteins with low recall rates suitable for determining oligomeric states with DCC-SMLM (*Figure 4*). The original DCC model (*Equations 1–3*) suggested a recall rate of only 0.12 for PAmCherry, but did not result in an optimal fit for our experimental data from reference proteins (*Figure 4A*). We reasoned that the function of the two fluorescent proteins may somehow become impaired in the context of fusion proteins. Therefore, we introduced a modification

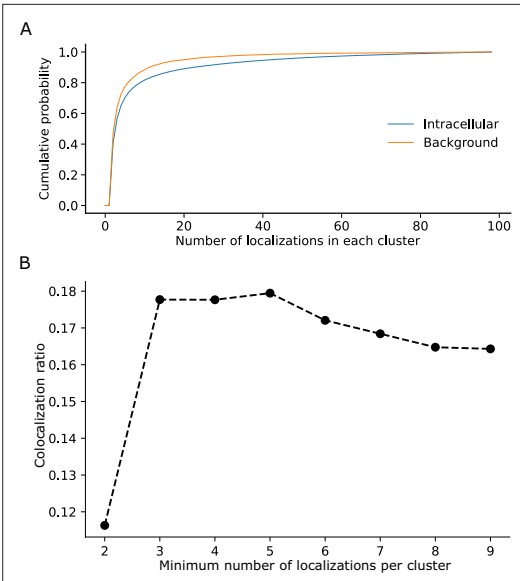

**Figure 5.** Determination of parameters for the clustering of localizations. (**A**) The cumulative probability distribution of the number of localizations per cluster in the background and within Flp-In T-Rex 293 cells expressing Barttin-mVenus-PAmCherry. While most clusters in the background only exhibit a small number of localizations, clusters within cells exhibit a higher proportion of larger clusters. (**B**) The dependence of the colocalization ratio p versus the lower cutoff to be considered a genuine cluster. A cutoff of 2 results in low ratios due to the inclusion of a large number of background red clusters, which interfere with the computation algorithm by excluding a large amount of colocalizations of the genuine clusters. While data on the stoichiometry in large protein clusters and macro-structures can be relevant, dual-color colocalization (DCC)-single molecule localization microscopy (SMLM) was developed to only resolve smaller oligomeric structures in situ as they are typically the smallest functional units of proteins. We therefore removed larger structures from further analysis. The number of clusters removed, however, was very low with typically more than 98% remaining (*Figure 1—figure supplement 1*).

The online version of this article includes the following figure supplement(s) for figure 5:

**Figure supplement 1.** Comparison of localization extraction using two different softwares: SMAP and SNSMIL.

factor, *m*, to describe the probability of both fluorescent proteins becoming dysfunctional. Fitting this modified DCC model to our reference data suggested a slightly higher recall rate of 0.17 for PAmCherry, and aligned well with the experimental data (*Figure 4A*). Previously reported recall rates for PAmCherry range from 0.04 when expressed in *Escherichia coli* (*Wang et al., 2014*) to about 0.5 in *Xenopus* oocytes (*Durisic et al., 2014*). This variability implies that the expression system may influence the functionality of fluorescent proteins. Nevertheless, it may also result from differences in image acquisition and data processing. The software SNSMIL (*Tang et al., 2015*), which was used in our study to extract localizations, is no longer optimal, and newer software such as SMAP (*Ries, 2020*) may be more sensitive in localization extraction and yield different recall rates. In comparison with SNSMIL, however, SMAP did not significantly improve the colocalization ratios because of its higher sensitivity to background signals (*Figure 5—figure supplement 1*). However, this may not be representative as SNSMIL was developed on our microscope and may thus perform better than expected and we recommend to use more up-to-date software. It should also be considered that many fluorescent proteins and dyes are sensitive to their specific microenvironment and it may not be taken for granted that the same values of *p* and *m* can be used in every circumstance. However, in our case, the same values gave reasonable results that matched the oligomeric states as determined by hrCNE (*Figure 4—figure supplement 2*).

Introduction of the modification factor *m* improved our original DCC model. Different from the factor *p*, which describes the recall rate of PAmCherry alone, *m* additionally regards those behaviors that lead to loss of fluorescence of both mVenus and PAmCherry simultaneously. This takes away the independence of the two signals and invalidates the use of *p* alone. Based on our experience with fluorescent protein-linked proteins, we assume that the most likely cause for such behavior is truncation within or proximal to mVenus. We often observe such events when solubilizing proteins containing fluorescent proteins that show up as smaller, additional bands on polyacrylamide gels such as seen in hrCNE

(*Figure 4—figure supplement 1*). We currently suspect that this is largely dependent on the linker sequence used. Therefore, we recommend to use the same linker for calibration proteins and POIs.

Another reason to carefully consider is the expression of endogenous proteins that can form oligomers with the transfected POIs. This would also reduce the apparent colocalization ratios due to the inclusion of subunits without fluorescent proteins. We used a publicly available database of protein expression (https://www.proteinatlas.org/) to make sure that our calibration proteins and POIs are not

expressed in the HEK293T cells that we used. The only protein with meaningful expression in this cell line was ClC-2 which is why we also used the bovine ClC-K channel as another dimer control. Colocalization ratios obtained for ClC-2, however, were similar to those of bClC-K (*Figure 4A*), indicating that dimerization with endogenous channels was negligible.

If the interest is only in determining higher oligomeric states (larger than dimers), the factor *m* may be omitted. However, as theoretically expected, the determination of a monomeric state becomes almost impossible without *m* (*Figure 4C–H*) when taking the uncertainty of all values into account. Alternatively, the determination of the most likely oligomeric state can be performed without the knowledge of either *p* or *m* using a comparison to the recorded colocalization ratios of reference proteins. We used the two-sample K.-S. test with Bonferroni-adjusted significance thresholds (*Figure 4I–N*). This has the potential to further simplify the determination of oligomeric states but is dependent on very accurate recordings of reference protein values.

To achieve the necessary spatial resolution across two colors, we implemented and compared simple methods of cell fixation of cells and post-imaging correction of common artifacts such as drift and chromatic aberrations. While our implementation is based on SMLM, it could also be adapted for other types of microscopy as stated above. In our experiments, there was no difference between a linear fit of chromatic aberrations and local average correction (*Figure 2* and *Figure 2—figure supplement 1*); however, this may depend on the quality of the optical setup and should be carefully evaluated for every instrument.

It should be emphasized that DCC-SMLM does not consider the behavior of individual protein complexes; instead, it is based on statistical information from a large number of protein complexes. Therefore, if the protein complex of interest does not exist in a single oligomeric state, DCC-SMLM counting can only report an average oligomeric state (simulated in *Figure 4—figure supplement 4*). Future efforts should aim to explore whether additional information obtained from DCC-SMLM experiments or other experimental designs could be used to increase its usefulness in analyzing proteins that adopt a mixture of oligomeric states.

To exclude an effect of expression density on the oligomeric state of our observed proteins, we used cells which covered a wide range of expression densities. A plot of the colocalization ratio versus the density of expressed clusters (*Figure 4—figure supplement 5*) did not reveal any significant correlation between the two values, effectively ruling out aggregation over the expression densities achieved in our samples. It would also be expected that higher expression densities may be used to decrease the variability of the results as the inclusion of more clusters per recordings would help the mean colocalization ratio per recording to fall closer to the actual mean.

Using DCC-SMLM, we determined the oligomeric state of proteins from the SLC26 and SLC17 families. Studies using Blue native electrophoresis (*Detro-Dassen et al., 2008*) and X-ray and cryo-EM (*Geertsma et al., 2015*; *Walter et al., 2019*) previously reported a dimeric assembly for SLC26, whereas microscopy approaches supported a tetrameric assembly (*Hallworth and Nichols, 2012*; *Sinha et al., 2010*). This discrepancy might indicate that SLC26 forms stable dimers that survive protein purification and solubilization as well as weakly associated 'dimer-of-dimer' tetramers in native membranes that may only be visible in intact cells. Although we cannot exclude the possibility that a minor fraction of prestin forms tetrameric structures, our results show that both prestin and SLC26A3 predominantly form dimers in intact HEK293T cells (*Figure 4B–D*).

We also studied the oligomeric state of vGlut1, -2, and -3 (Slc17a7, Slc17a6, and Slc17a8, respectively), and sialin (SLC17A5) and found that they are likely to be monomeric. Again, these results are consistent with native gel electrophoresis (*Figure 4—figure supplement 2*). An earlier report suggested dimeric organization for SLC17 transporters based on SDS-PAGE results (*Andersen, 2004*) but a recent cryo-EM study reported a monomeric structure (*Li et al., 2020*). Functional expression studies have also demonstrated that vesicular glutamate transporters and sialin can function without accessory subunits (*Eriksen et al., 2016*; *Morin et al., 2004*), which, when considered alongside our results, confirms that SLC17 proteins are functional monomers.

Our study provides a new approach to determine the oligomeric state of membrane proteins in situ. DCC-SMLM overcomes many of the drawbacks and requirements of previous methods, making determination of oligomeric states more reliable and much easier to perform. In fact, while we use the *x*- and *y*-coordinates provided by SMLM, the DCC principle is not reliant on SMLM and the algorithm may also be used with other microscopy modalities given enough spatial resolution to separate

individual protein complexes. Examples may include structured illumination (*Kner et al., 2009*) and super-resolution radial fluctuations (*Gustafsson et al., 2016*) which are otherwise difficult to quantify at the single molecule level because they rely on the mathematical processing of multiple raw images to yield super-resolution images removing much of the information required by other algorithms.

# Materials and methods

**Key resources table**

| Reagent type (species) or resource | Designation | Source or reference | Identifiers | Additional information |
|---|---|---|---|---|
| Cell line (*Homo sapiens*) | tsa201 (HEK293T) | ECACC via Sigma Aldrich | Cat# 96121229-1VL | |
| Cell line (*Homo sapiens*) | Flp-In T-Rex 293 | Thermo Fisher Scientific | Cat# R78007 | |
| Transfected construct (*Homo sapiens*) | SLC26A3 | 10.1074/jbc.M704924200 | GenBank:NM_000111 | |
| Transfected construct (*Rattus norvegicus*) | Slc26a5 (Prestin) | 10.1074/jbc.M704924200 | GenBank:NM_030840 | |
| Transfected construct (*Homo sapiens*) | ClC-2 | 10.1093/hmg/4.3.407 | Swiss-Prot:P51788 | With Y17H and R210H inserted to conform to 360 control chromosomes (*Haug et al., 2003*) |
| Transfected construct (*Bos taurus*) | bClC-K | This paper | | Cloned from bovine kidney tissue |
| Transfected construct (*Homo sapiens*) | EAAT2 | 10.1074/jbc.M110.187492 | | |
| Transfected construct (*Mus musculus*) | vGlut1 | Other | | From Dr R Guzman, FZ Jülich. Carries a mutation of the di-leucin motif to alanine to facilitate plasma membrane insertion (see *Eriksen et al., 2016*) |
| Transfected construct (*Mus musculus*) | vGlut2 | Other | | From Dr R Guzman, FZ Jülich. Carries a mutation of the di-leucin motif to alanine to facilitate plasma membrane insertion (see *Eriksen et al., 2016*) |
| Transfected construct (*Mus musculus*) | vGlut3 | Other | | From Dr R Guzman, FZ Jülich. Carries a mutation of the di-leucin motif to alanine to facilitate plasma membrane insertion (see *Eriksen et al., 2016*) |
| Transfected construct (*Mus musculus*) | Sialin | Other | | From Dr R Guzman, FZ Jülich. Carries a mutation of the di-leucin motif to alanine to facilitate plasma membrane insertion (see *Eriksen et al., 2016*) |
| Transfected construct (*Homo sapiens*) | Barttin | 10.1097/HJH.0b013e3283140c9e | GenBank:NM_057176 | Obtained via Dr AL George |
| Transfected construct (*Homo sapiens*) | $K_{ir}$2.1 | 10.1073/pnas.102609499 | | Obtained via Prof. Dr J Daut |
| Other | TetraSpeck fluorescent beads (100 nm) | Thermo Fisher Scientific | Cat# T7279 | Fluorescent microspheres used for the correction of drift and chromatic aberration, as well as for the general calibration of the microscope |

## Cell culture and DNA constructs

HEK293T cells and Flp-In T-REx 293 cells were grown in DH10 medium (DMEM supplemented with 50 U/ml penicillin-streptomycin and 10% FBS at 37 °C and 5% $CO_2$; *Tan et al., 2017*). For Flp-In T-REx 293 cells, the medium was additionally supplemented with 10 µg/ml blasticidin S HCl and 100 µg/ml hygromycin B. None of the cell lines used were authenticated. However, no commonly misidentified lines were used. Cell lines were tested for mycoplasma contamination twice per year. cDNAs encoding the fusion proteins were inserted into the cloning site of the pcDNA5/FRT/TO or pcDNA3 vector plasmid via overlap extension PCR, restriction digestion, and DNA ligation.

## Cleaning of coverslips

Piranha solution was made by slowly adding 1 volume of hydrogen peroxide solution (30%, Merck) into 3 volumes of concentrated sulfuric acid (> 95%, Sigma-Aldrich) while stirring. Caution: Piranha solution reacts violently with organic matter and must be handled and disposed of according to institutional guidelines! Coverslips (VWR) were placed in a glass container and freshly prepared piranha solution was added. Following incubation for 4–7 days at room temperature, coverslips were rinsed with double distilled water at least 20 times. Afterward, coverslips were stored in double distilled water at 4 °C for experiments within the next 2 weeks.

## Preparation of cells for SMLM imaging

For SMLM imaging, HEK293T cells were sparsely seeded onto clean 25 mm coverslips in 1.5 ml DH10 in 3.5 cm Petri dishes 2–3 hr before transfection. Cells were transfected with 0.5–1.0 μg plasmid DNA plus 1.0 μl Lipofectamine 2000 (Life Technologies) according to the manufacturer's instructions and cultured overnight (12–16 hr) before fixation. Flp-In T-REx 293 cells were seeded on coverslips in the same way and 0.01 μg/mL tetracycline was added to the culture medium 14.5 hr before fixation.

Afterwards, cells were washed three times with 2 ml, 4 °C PBS and fixed in 1 ml of –20 °C cold methanol (Gradient grade for liquid chromatography, Merck) for 5 min at –20 °C. Methanol was then removed and the cells were washed five times with 4 °C cold PBS.

For experiments using aldehyde fixation, 4 g PFA (Merck) was dissolved in 80 ml PBS heated to 60 °C. Once completely dissolved, the solution was filled to 100 ml with PBS. One ml aliquots were stored at –20 °C for long-term storage. Prior to fixation, aliquots were warmed to 37 °C and vortexed to dissolve precipitates. Eight μl of a 25% GA solution (Electron microscopy grade, Agra Scientific) was added to the PFA aliquot for a final concentration of 0.2% (w/v) GA. Cells were washed three times with 37 °C warm PBS and fixed in 1 ml of the PFA/GA solution for 30 min at 37 °C. Afterward, cells were washed three times with PBS, followed by incubation in 0.5% (w/v in PBS) ice-cold $NaBH_4$ (Sigma-Aldrich) solution for 1 hr at 4 °C. Fixed cells were stored in 2 ml PBS in the dark at 4 °C until imaging within 24 hr.

## Preparation of bead samples for imaging

To prepare bead samples, 5 μl multi-color TetraSpeck fluorescent bead solution (100 nm diameter, Thermo Fisher Scientific) was diluted with 400 μl double distilled water, added to cleaned coverslips and let dry.

## SMLM imaging

Imaging was performed on a custom-built wide field and total internal reflection fluorescence microscope described previously (*Stölting et al., 2015b*; *Tang et al., 2015*). We used 561 and 405 nm diode lasers for imaging and photoactivation of PAmCherry, respectively. Prior to imaging, the sample was washed three times with PBS and mounted in the chamber. One μl TetraSpeck fluorescent bead solution was diluted in 0.5 ml PBS, added to the coverslip, and incubated for 1–2 hr in the dark. Afterward, the solution was removed and 1 ml PBS was added for imaging. To reduce vaporization, the chamber was covered.

The focal plane was adjusted using red color signals of the fluorescent beads on the coverslip surface. A transmission image was taken to record positions of cells and beads. Videos were recorded at an exposure of 85.59 ms per frame, during which the illumination lasted for 50 ms. Background signals were bleached by illumination with the 561 nm laser at full power for 500–1000 frames before the recording started. Color channels were recorded sequentially, starting with the green channel (for mVenus), followed by the red channel (for PAmCherry). See *Table 3* for details about the laser power and filter settings. The intensity of the 405 nm laser used for photoactivation of PAmCherry was gradually increased during the recording to keep signals sparsely distributed over time. The recording was stopped when no signals appeared. After recording both color channels, the focus was re-adjusted using the fluorescent beads, and the displacement of the sample stage was taken as the axial drift. Recordings with an axial drift > 300 nm were discarded.

## Extraction of SMLM signals

Raw videos were processed by SNSMIL (*Tang et al., 2015*) with the following parameters: optical magnification, 200; numerical aperture, 1.49; gain, 300; e/AD count, 11.9; excess noise factor square, 2; pixel diameter, 16 µm; bias offset (AD counts), 100; quality threshold, 1; PSF width tolerance, 3; fitting model, fixed SD 2D Gaussian non-integral model. For recordings of the green channel (mVenus and beads), the peak emission wavelength was set to 528 nm and for the red channel (PAmCherry and beads) to 595 nm. Coordinates obtained from SNSMIL were translated from pixels to nm using the known size of 80 × 80 nm/pixel. Binary reconstructions were plotted using a custom written python script using the package Pillow (v.6.0.0). The same parameter settings were used when SMAP was used for data extraction.

## Drift correction

The algorithm was adapted from a previously reported application using nanodiamonds (*Yi et al., 2016*). First, the drift was corrected relative to the first frame of each color. Beads close to the recorded cells were manually selected from the reconstructed image using Fiji (*Schindelin et al., 2012*) and precise coordinates were extracted from the output of SNSMIL using a custom Python script. In a recording with $n$ total frames, the average drift ($\delta_k$) on the $k^{th}$ frame ($k$ = 1, 2, 3, …, $n$) was calculated across all $m$ beads (with the index $i$ = 1, …, $m$) relative to the first frame as:

$$\delta_k = \frac{\sum\limits_{i=1}^{m}\left(x_{i,k}-x_{i,1}, y_{i,k}-y_{i,1}\right)}{m_k} \tag{7}$$

The drift was subtracted from each localization ($p_k$) to give the drift corrected position for each frame ($p'_k$):

$$p'_k = p_k - \delta_k \tag{8}$$

As the red channel was recorded after the green channel, the corrected positions of the red channel were further corrected relative to the last frame in the green channel:

$$p''_{\text{Red}} = p'_{\text{Red}} - \delta_{\text{Green},n} \tag{9}$$

## Determination of ACA

To determine ACA, a bead sample was used. First, the focus was set in the red channel. The stage was moved from 700 nm above to 700 nm below the focal plane in 100 nm steps and 10 frames in each color channel were recorded at each step. The mean intensities of the signals from each step were then analyzed.

## Correction of LCA

For the determination of LCA, multiple beads (> 10 beads) without cells were recorded in wide field mode for 50 frames in the green and red channel, respectively. Data from multiple recordings were concatenated into a single dataset (100–200 beads in total, evenly distributed around the field of view) for each color channel. In this dataset, the coordinates of signals were clustered using the DBSCAN implementation of the python package scikit-learn (v.0.21.0) (*Pedregosa et al., 2011*) to assign localizations to individual beads. The parameter *eps* defines the upper threshold of the Euclidean distances between any two localizations for a single bead and was empirically determined as 40 nm. The parameter *min samples* defines the lower threshold of localizations required for each cluster and was set to 50 to ensure that localizations must occur in every frame of the recording. Following clustering in separate color channels, distances to the nearest neighboring cluster in the other color channel were calculated and considered to be from the same bead if within 200 nm. The second (red) was corrected toward the first recorded color (green). For every bead in the dataset, the LCA was determined as the shift between the mean coordinates of the clustered localizations in the red ($p_{\text{Red}}$) and green ($p_{\text{Green}}$) channels:

$$A = p_{\text{Red}} - p_{\text{Green}} \tag{10}$$

In general, correction of LCA was performed relative to the green channel according to yield corrected coordinates for each localization:

$$p'''_{\text{Red}} = p''_{\text{Red}} - A \tag{11}$$

For the regional LCA correction (rCA) (*Churchman and Spudich, 2012*; *Georgieva et al., 2016*), the field of view was divided into 100 squares (each 1049 × 1049 nm²). Within each square with the coordinates ($i,j$), the mean LCA ($A_{i,j}$) was calculated by averaging the LCA from the individual beads found within this square. For squares without any beads, $A_{i,j}$ was interpolated from neighboring regions. rCA for samples of interest was performed by subtracting $A_{i,j}$ from the localizations in each square according to *Equation 11*.

To correct LCA by linear fit (fCA) (*Erdelyi et al., 2013*; *Kozubek and Matula, 2000*), the LCA ($A_x$, $A_y$) was determined as described above along the positions of all beads detected in the red channel ($x$, $y$). The LCA varies across the field of view according to:

$$\begin{pmatrix} A_x \\ A_y \end{pmatrix} = K \begin{pmatrix} x - x_0 \\ y - y_0 \end{pmatrix} \tag{12}$$

$K$, $x_0$, and $y_0$ were determined by a fit of the LCA and positions of all beads present in the sample. With these values, the LCA ($A_s$) of any given localization with position ($x_s$, $y_s$) in samples of interest was then calculated by

$$A_s = K \begin{pmatrix} x_s - x_0 \\ y_s - y_0 \end{pmatrix} \tag{13}$$

LCA correction was then performed according to *Equation 11*.

## Cluster analysis of mVenus and PAmCherry localizations

One emission event from either mVenus or PAmCherry lasts for several frames, generating a series of localizations that are temporally and spatially close (*Durisic et al., 2014*; *Nan et al., 2013*). Similar to the assignment of localizations to individual beads, we used DBSCAN for a cluster analysis of localizations. The parameter 'eps' was empirically determined as 40 nm while 'min samples' was set to 10 for mVenus and 6 for PAmCherry (*Figure 5*). Non-clustered signals were discarded from further analysis.

## Pair correlation analysis and computation of colocalization ratios

Mean coordinates of localization clusters for each color channel were used to represent the positions of the fluorescent proteins. To determine the cutoff value of the inter-color distance for colocalization, we performed pair correlation analysis across both color channels (*Sengupta et al., 2011*; *Sengupta and Lippincott-Schwartz, 2012*). In short, the densities of PAmCherry localization clusters around any given mVenus cluster was calculated according to

$$g_{\text{MF}}(r) = \frac{a}{\pi N_M N_F} \sum_{i=1}^{N_M} \frac{dN_{Fi}(r)}{(2r+dr)\,dr} \tag{14}$$

in which $a$ is the area of the selected ROI; $N_M$ and $N_F$ are the numbers of green and red clusters in the ROI, respectively; $dN_{Fi}(r)$ represents the number of red clusters that fall into the annulus around the $i^{\text{th}}$ cluster with the inner radius $r$ and the outer radius $r + dr$. To accelerate computing, distances to all PAmCherry clusters within 400 nm from each mVenus cluster were calculated using the spatial. distance.cdist function from the python package SciPy v.1.2.1 (*Virtanen et al., 2020*). The values were grouped into bins of 20 nm width and the pair correlation calculated for each bin. The correlation length from the pair correlation analysis was used as the maximum colocalization distance between clusters (100 nm; *Figure 2H*). Colocalization was registered for an mVenus cluster if a single red cluster was found in a distance shorter than this maximum colocalization distance. The numbers of clusters and colocalizations were counted in cellular and background regions and applied to *Equations 5 and 6* to calculate colocalization ratios.

In addition, to prevent miscounting due to protein aggregation, those clusters with a diameter larger than 500 nm were excluded from counting. The radius of a cluster was determined as $\mu + 2\sigma$ of the Euclidean distances of all localizations to the center (mean coordinates) of the cluster, as shown *Figure 2B*.

## Estimation of the oligomeric state of the POI

To estimate the oligomeric state of POIs and the observed colocalization ratios, we defined the coefficient of mismatch, $M^2(n)$ as

$$M^2(n) \equiv 1 - \frac{\sum_i (x_i - \bar{x})^2}{\sum_i [x_i - E(n)]^2} \tag{15}$$

in which $x$ is the mean of all observations ($x_i$) and $E(n)$ is the expected colocalization ratio for the oligomeric state $n$. The expected value was calculated from *Equation 4* with the optimal values of $m$ and $p$, or from *Equation 1* with the optimal $p$ alone, as determined by the respective fitting to the calibration data (*Figure 4A*). For the bootstrapping analysis to determine 95% confidence intervals of the mean, $M^2$ was calculated from resamples of the colocalization ratios measured from each POI and the reference proteins. The Python module scipy.stats.ks_2samp was used to perform the two-sample K.-S. test.

## Fitting of the DCC model to the experimental data

$m$ and $p$ were determined as the fit parameters of *Equation 4* to the experimental data. The value pair ($m$, $p$), with $m$ ranging from 0 to 1 and $p$ from 0 to 0.4, both with a 0.01 increment, was used to calculate the theoretical colocalization ratios for oligomeric states $n$ = 1, 2, 3, and 4. The calculated ratios were then compared with the experimental values obtained from all the proteins with known oligomeric states to calculate the coefficient of determination ($R^2$). In the case when $m$ was not included, the fitting method was similar, but with $m$ left out.

## High-resolution clear native electrophoresis

HEK293T cells were transfected with 3–5 µg of plasmid DNA using the calcium phosphate method and treated as described previously (*Guzman et al., 2017*). In brief, cells were washed with ice-cold PBS and incubated for 15 min with a lysis buffer containing 0.1 M sodium phosphate, either 0.5% (w/v) digitonin or 20 mM DDM, protease inhibitors, and 20 mM iodoacetamide. The buffer was then transferred into a 1.5 ml Eppendorf tube. After centrifugation at 4 °C, an aliquot of the supernatant containing approximately 10 µg of protein was loaded into a native gel. The 4–14% acrylamide gradient gels were prepared as described (*Wittig et al., 2007*). The anode buffer contained 25 mM imidazole/HCl, pH 7.0, and the cathode buffer contained 50 mM tricine, 7.5 mM imidazole/HCl, pH 7.0 supplemented with the anionic detergent DOC (0.05%), and the non-ionic detergent Triton X-100 (0.05%) (*Wittig et al., 2007*). The gels were run in a cold room (4 °C) and the starting voltage was set to 100 V. After 1 hr, the voltage was increased to 150 V for another 2 hr. Gels were scanned on a fluorescence gel scanner (Typhoon FLA 9500, GE Healthcare, Freiburg, Germany) at 100 µm resolution. mVenus was excited with the built-in 473 nm laser and the emission was filtered with a 530/20 bandpass filter. Gel images were visualized using Fiji (*Schindelin et al., 2012*).

## Acknowledgements

This research was supported by the Deutsche Forschungsgemeinschaft (FA 301/15–1 to Ch.F.) as part of Research Unit FOR 5046, project P4. We are grateful to Dr Iman Abdollahzadeh, Dr Johnny Hendriks, and Dr Thomas Gensch for their generous help with the use of the imaging setup. Many thanks also go to Cora Hannack, Dr Raul E Guzman, and Arne Franzen for providing plasmids encoding vGlut and sialin, as well as to Dr Maddalena Comini for providing help while cloning bClC-K. We also thank Ashley Craig of Alchemy Editorial Services Ltd for language editing of the initial manuscript.

## Additional information

### Funding

| Funder | Grant reference number | Author |
| --- | --- | --- |
| Deutsche Forschungsgemeinschaft | FA 301/15-1 | Christoph Fahlke |

The funders had no role in study design, data collection and interpretation, or the decision to submit the work for publication.

### Author contributions

Hua Leonhard Tan, Conceptualization, Data curation, Software, Formal analysis, Validation, Investigation, Visualization, Methodology, Writing - original draft, Writing – review and editing; Stefanie Bungert-Plümke, Formal analysis, Investigation, Visualization, Writing – review and editing; Daniel Kortzak, Formal analysis, Methodology, Writing – review and editing; Christoph Fahlke, Conceptualization, Supervision, Funding acquisition, Writing – review and editing; Gabriel Stölting, Conceptualization, Data curation, Software, Formal analysis, Supervision, Investigation, Visualization, Methodology, Project administration, Writing – review and editing

### Author ORCIDs

Hua Leonhard Tan (ID) http://orcid.org/0000-0002-3938-3454
Stefanie Bungert-Plümke (ID) http://orcid.org/0000-0002-4650-3274
Gabriel Stölting (ID) http://orcid.org/0000-0002-2339-0545

### Decision letter and Author response

Decision letter https://doi.org/10.7554/eLife.76631.sa1
Author response https://doi.org/10.7554/eLife.76631.sa2

## Additional files

### Supplementary files

• Transparent reporting form

### Data availability

The datasets including the extracted localization are available for download via https://doi.org/10.5281/zenodo.6012450. Raw video files can only be provided upon request due to their large file sizes. We implemented a version of DCC-SMLM as a python library on GitHub (https://www.github.com/GabStoelting/DCC-SMLM, copy archived at swh:1:rev:246cd479244115bd9bcf00ab04caf2490430fb1d).

The following dataset was generated:

| Author(s) | Year | Dataset title | Dataset URL | Database and Identifier |
| --- | --- | --- | --- | --- |
| Tan HL, Bungert-Plümke S, Kortzak D, Stölting G, Fahlke CH | 2022 | Determination of protein stoichiometries via dual-color colocalization with single molecule localization microscopy | https://doi.org/10.5281/zenodo.6012450 | Zenodo, 10.5281/zenodo.6012450 |

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
