## [Editor Report]

The authors present a method for measuring the average oligomerization state of fluorescently tagged membrane proteins by single-molecule localization microscopy (SMLM). In contrast to many other SMLM methods which aim to count subunits in membrane protein complexes, the authors aim to deduce the average oligomerization state from the probabilistic co-detection of at least 1 'reporter' fluorophore, which has relatively poor detection efficiency, with the detection of at least 1 fused 'marker' fluorophore. They calibrate the method against a set of proteins with known oligomerization states (validated against high-resolution clear native gel electrophoresis) and then apply it to convincingly clarify the oligomerization state of SLC26 and SLC17 family member membrane proteins. Although the approach is limited to measurements of the average oligomerization state, and as such is not suitable to measure a distribution of (higher) oligomerization states, it is nonetheless potentially very useful for identifying oligomerization states of unknown proteins in native cells, and furthermore works well with fluorophores that have poor detection efficiencies. The provided software should be sufficient to allow other researchers with some experience in Python to perform this analysis on their own data.

---

## [Decision Letter]

**Decision letter after peer review:**

Thank you for submitting your article "Determination of protein stoichiometries via dual-color colocalization with single molecule localization microscopy" for consideration by *eLife*. Your article has been reviewed by 2 peer reviewers, including Marcel P Goldschen-Ohm as Reviewing Editor and Reviewer #1, and the evaluation has been overseen Richard Aldrich as the Senior Editor.

Essential revisions:

Please address all of the comments from both reviewers. In particular, pay close attention to the following points:

1) On the theory side, a rigorous probabilistic framework for the assignment of the most likely oligomerisation state is missing. This includes a sensitivity analysis of Equation 1 (or, better, of Equation 6) which highlights at which n or p this method is most sensitive. Also, no confidence intervals for fitted values of m and p were provided which could be used in such a sensitivity analysis. For example, it is clear that a high detection efficiency p renders the method insensitive. The optimal range p = 0.2-0.4 mentioned in the discussion is not substantiated. Also, the uncertainty of the experimentally determined values p and m could be accounted for by error propagation.

2) The SMLM detection/processing details are not state-of-the-art (PSF fit with fixed SD 2D Gaussian; not using maximum likelihood estimation for fitting; DBSCAN algorithm to group raw (single-frame) data). In conjunction with setting a minimum value of 6 (PAmCherry) and 10 (mVenus) for the number of localisations per cluster, these together might contribute to the poor detection efficiency for PAmCherry of 0.12, which is in contrast to the reported maturation efficiency of the protein, and which the authors attribute to protein misfolding. The detection, localisation and grouping of fluorescent events could be substantially improved by using maximum likelihood fitting of experimental point spread functions and post-filtering according to the log-likelihood ratio (LLR), as e.g. offered by the open source software SMAP (Ries, 2020, Nature Methods). This is expected to improve the detection of short fluorophore blinks while improving the rejection of background events. This may also impact the large variation observed from cell to cell, the limitations/requirements of which should be discussed.

3) The manuscript would also benefit from more discussion regarding the origin of the factor 'm'. Do HEK293T cells natively express any untagged protein corresponding to the transfected POIs? If not, it would be important to state this explicitly in the text. If yes, this violates the assumption of the analysis; in fact, it would be expected to contribute to the 'm' factor in equation 6 and lead to a significant variation of this factor from cell to cell, depending on the relative expression levels of tagged and untagged protein. Thus, a knock-out cell line needs first to be created before introducing the tagged POI. Also, it is unlikely that the author's attribution of the very low detection efficiency to a 'misfolded' fraction of proteins is the only possible explanation. For example, the coexistence of different oligomerisation states is expected to have a similar effect than terminated translation. This could be systematically explored by computer simulations to better justify the introduction of this factor and the limitations this implies for the calibration of the method. Finally, is Equation 6 even necessary to determine the oligomerization state?

4) The word stoichiometry in the title of the paper is misleading. Although the technique could be applied to measure the oligomerisation state of different subunits in independent samples using different expression constructs, and thus an average stoichiometry could be determined, it is not suitable to directly measure stoichiometries of different subunits in the same sample.

5) An open-source software tool would find wider-spread application and complement existing methods to measure the oligomerisation state of membrane proteins from monomers to tetramers using relatively standard PALM approaches.

6) Some of the semantics need to be better defined for a more general readership.

7) Regarding the fits to the data in Figure 4A, I assume that all of the (e.g., red) points were globally fit to the data for all 5 proteins with known stoichiometries using a single value of p. However, this is not stated in the text, which made it initially difficult for me to understand what the authors were doing here. Please describe the fitting in more detail in the text.

*Reviewer #1 (Recommendations for the authors):*

1. I am under the impression that the approach requires identification of distinct proteins, hence the need for SMLM. The authors state that they achieve a 30nm radial resolution, so I assume that an inherent assumption here is that multiple proteins within 30nm of each other must be assumed to be rare? Even assuming this is ok, a 100nm cutoff is used for determining colocalization of spots in the two color channels. Why so much larger than 30nm? How do the authors ensure that only one protein is within each of these 100nm spots? Or can multiple PAmCherry spots be colocalized with the same mVenus spot, or vice-versa? This seems confusing to me. Or if this does not matter, please explain as the theory (e.g., Equation 3) requires counts of numbers of proteins. In the discussion, the authors suggest that their approach is actually not reliant on SMLM at all, and only requires "enough spatial resolution". Please define what "enough" means. And again, if resolving individual proteins is actually not required here, then this needs to be clarified.

2. Some of the semantics need to be better defined for a more general readership. What is a "localization"? Observation of a single fluorophore on a single frame, or the identified location of a single fluorophore across frames? Does a cluster of localizations represent a single protein, or a cluster of proteins? If it is the former, then requiring at least 3 localizations for a cluster to be analyzed (e.g., see Figure 5B) may limit background noise, but would also remove proteins where PAmCherry bleached within a couple of frames. What is the distribution of bleach times for PAmCherry, and what fraction are discarded by this cutoff? The authors suggest that this cutoff removes PAmCherry localizations that are likely to be background noise which do not colocalize with mVenus clusters. But if they do not colocalize, then how do they affect the computation at all, as I thought only colocalized clusters were considered? Overall, the methods should be described in more detail for a general readership.

3. I have some reservations regarding the use of Equation 6, which to some extent appears to be a fudge factor given that the data does not quite fit the initial simple theory (Equation 1). First, it is my understanding that p encompasses all of the things that can lead to a fluorophore not being observed. So why then are some ways in which this could occur such as misfolding or truncation treated separately? Second, Equation 6 needs to be better explained as to why it is appropriate to describe misfolding or truncation events. Third, I would like to see Figure 4C-N repeated without using the fudge factor m. Is Equation 6 really needed to reliably determine the stoichiometries of the tested exchangers/transporters? And lastly, why should we expect m to be the same for different proteins? It seems to me that misfolding or truncation may be highly protein dependent. If m does differ from protein to protein, then it seems like this entire approach is no longer robust, at least for monomers.

4. Equation 4 is introduced as a means to limit the contribution of background noise, but thereafter it appears that the authors just apply Equation 3 to their data. If so, what is the point of Equation 4, or is this a mistake in the text? Also, the variable N_CO in Equation 4 seems to be the same thing as N_MF in Equation 3? If so, please stick to one or the other, and if not please clarify.

5. Regarding the fits to the data in Figure 4A, I assume that all of the (e.g., red) points were globally fit to the data for all 5 proteins with known stoichiometries using a single value of p. However, this is not stated in the text, which made it initially difficult for me to understand what the authors were doing here. Please describe the fitting in more detail in the text.

6. The blue points in Figure 4A are hard to distinguish from the black points. Please choose a better representation. Also, for Figure 4A,B please show all of the data points rather than just box plots plus outliers. There are not so many data points to make this unreasonable in this case.

*Reviewer #2 (Recommendations for the authors):*

The detection, localisation and grouping of fluorescent events could be substantially improved by using maximum likelihood fitting of experimental point spread functions and post-filtering according to the log-likelihood ratio (LLR), as e.g. offered by the open source software SMAP (Ries, 2020, Nature Methods). This is expected to improve the detection of short fluorophore blinks while improving the rejection of background events.

The full potential, but also the limitations of the approach do not become clear because Equation 1 (or Equation 6) are not rigorously analysed with respect to their sensitivity/ability to differentiate different oligomerisation states. For example, it is clear that a high detection efficiency p renders the method insensitive. The optimal range p = 0.2-0.4 mentioned in the discussion is not substantiated. Also, the uncertainty of the experimentally determined values p and m could be accounted for by error propagation.

The manuscript would also benefit from more discussion regarding

– The origin of the factor 'm'. Do HEK293T cells natively express any untagged protein corresponding to the transfected POIs? If not, it would be important to state this explicitly in the text. If yes, this violates the assumption of the analysis; in fact, it would be expected to contribute to the 'm' factor in equation 6 and lead to a significant variation of this factor from cell to cell, depending on the relative expression levels of tagged and untagged protein. Thus, a knock-out cell line needs first to be created before introducing the tagged POI.

– The large variation observed from cell to cell. Figure 4, A and B, as well as Figure 4 —figure supplement 4 show substantial variation of the colocalization ratio P measured from cell to cell, as large as ranging from 0.15 to 0.45 (EAAT2) or 0.1 to 0.4 (CIC-2). The methods do not state how large the imaged/analysed FOV in a single cell was. According to Figure 1B, it was at least 5x5 µm^2^. With 1-2 clusters per µm^2^, this corresponds to 25-50 clusters. Together with the low detection efficiency p=0.12 for PAmCherry, this is expected to result in a substantial variability of P from measurement to measurement. This underlines how an improved detection efficiency, as could be achieved when using state-of-the-art SMLM, might translate in more accurate measurements.

---

## [Author Response]

Essential revisions:Please address all of the comments from both reviewers. In particular, pay close attention to the following points:1) On the theory side, a rigorous probabilistic framework for the assignment of the most likely oligomerisation state is missing. This includes a sensitivity analysis of Equation 1 (or, better, of Equation 6) which highlights at which n or p this method is most sensitive. Also, no confidence intervals for fitted values of m and p were provided which could be used in such a sensitivity analysis. For example, it is clear that a high detection efficiency p renders the method insensitive. The optimal range p = 0.2-0.4 mentioned in the discussion is not substantiated. Also, the uncertainty of the experimentally determined values p and m could be accounted for by error propagation.

For our sensitivity analysis of the optimal range of *‘m’* and *‘p’*, we created 2D plots, which show the optimal values of these two parameters to separate the indicated oligomeric states (monomer through tetramer) from the next closest oligomeric states (now shown in Figure 1 —figure supplement 1). These plots substantiate our statement that a value of *‘p’* between 0.2 – 0.4 usually allows for the best separation of oligomeric states. Furthermore, the plot demonstrates that *‘m’* should usually be smaller than 0.3, a finding which we previously did not describe. It must be noted that this analysis is based on the theoretical considerations of Equation 4 (The result at *m*=0 is equivalent to the result of Equation 1). The results are shown as normalized differences. Actual analyses of microscopy recordings include some degree of technical noise and biological variability which also impair the analysis of such samples.

Therefore, using a bootstrap resampling algorithm, we now determine the uncertainty of the values of *‘p’* and *‘m’* calculated from the observed colocalization ratios of our calibration proteins. We also used bootstrap resampling to determine the combined uncertainty of the “coefficient of mismatch” (COM) resulting from the error propagation of the uncertainties of *‘p’* and *‘m’* in addition to the uncertainty of the measured colocalization ratios of proteins of interest. This allows us to determine and evaluate the most likely oligomeric state based on a comparison of 95% confidence intervals (new Figure 4C-H). We changed the text to include the new analysis.

We agree that understanding *‘m’* and *‘p’*, as well as exploring the impact of error propagation, is important. However, we would like to point out that it is possible to also apply our method in a model-free strategy solely based on a comparison of measured colocalization ratios of proteins of interest (with unknown oligomeric states) with recorded values from reference proteins. This was already previously performed using the Kolmogorov-Smirnov test in Figure 4I-N and gave identical results. We now better highlight this aspect .

2) The SMLM detection/processing details are not state-of-the-art (PSF fit with fixed SD 2D Gaussian; not using maximum likelihood estimation for fitting; DBSCAN algorithm to group raw (single-frame) data). In conjunction with setting a minimum value of 6 (PAmCherry) and 10 (mVenus) for the number of localisations per cluster, these together might contribute to the poor detection efficiency for PAmCherry of 0.12, which is in contrast to the reported maturation efficiency of the protein, and which the authors attribute to protein misfolding. The detection, localisation and grouping of fluorescent events could be substantially improved by using maximum likelihood fitting of experimental point spread functions and post-filtering according to the log-likelihood ratio (LLR), as e.g. offered by the open source software SMAP (Ries, 2020, Nature Methods). This is expected to improve the detection of short fluorophore blinks while improving the rejection of background events. This may also impact the large variation observed from cell to cell, the limitations/requirements of which should be discussed.

To investigate whether SMAP can improve the recall rate, we reanalyzed our recordings with barttin. This protein is monomeric and the observed colocalization ratios should hence be directly proportional to the recall rate of PAmCherry. As shown in the new Figure 5 —figure supplement 1, SMAP detected more localizations and clusters in intracellular and background regions (Figure 5 - figure supplement 1A-D), resulting in similar colocalization ratios (Figure 5 —figure supplement 1E). While SNSMIL is clearly based on outdated concepts, it was developed using the very same microscope setup used for our study, which may help it to perform better than expected . We agree that newer software should be used for future applications of DCC-SMLM and already adapted our open-source implementation of the algorithm to accept files generated using SMAP. We changed the discussion on Page 14 to better explain the interdependence on the localization-extraction software used.

We already tested a wide range of minimum numbers of localizations per cluster (2 to 9, Figure 5B) but did not observe improved colocalization ratios of mVenus with PAmCherry clusters. Low values even decreased the ratio, likely because of the inclusion of random colocalization with noise. We now expand our explanations regarding this issue on Page 13 and the legend to Figure 5.

3) The manuscript would also benefit from more discussion regarding the origin of the factor 'm'. Do HEK293T cells natively express any untagged protein corresponding to the transfected POIs? If not, it would be important to state this explicitly in the text. If yes, this violates the assumption of the analysis; in fact, it would be expected to contribute to the 'm' factor in equation 4 and lead to a significant variation of this factor from cell to cell, depending on the relative expression levels of tagged and untagged protein. Thus, a knock-out cell line needs first to be created before introducing the tagged POI. Also, it is unlikely that the author's attribution of the very low detection efficiency to a 'misfolded' fraction of proteins is the only possible explanation. For example, the coexistence of different oligomerisation states is expected to have a similar effect than terminated translation. This could be systematically explored by computer simulations to better justify the introduction of this factor and the limitations this implies for the calibration of the method. Finally, is Equation 6 even necessary to determine the oligomerization state?

Data available from the Human Protein Atlas (https://proteinatlas.org) indicate negligible expression of most of our reference proteins in HEK293T cells. Only the ubiquitous chloride channel ClC-2 is expressed in these cells, and we therefore used ClC-K channels as an additional dimer reference. We found comparable colocalization ratios for ClC-2 and ClC-K channels, suggesting negligible dimerization with endogenous channel subunits. This information is now given in the Discussion section.

To better explain the need to include the factor ‘*m*’ in our analysis – and in response to a comment by reviewer #2 – we moved the introduction of the factor *m* to the theoretical considerations at the beginning of the Results section of the revised manuscript. The equations with and without *m* are later compared, based on our experimental observations . In general, the simpler model (Equation 1, based only on *‘p’*) worked well for higher oligomeric states (dimer and upwards), whereas the observed ratios for monomers were consistently higher than predicted. There is neither meaningful expression of endogenous, un-tagged subunits (see above) nor of other family members. In biochemical analyses we often observe cleavage of fluorescent proteins from the ends of proteins of interest (also visible in Figure 4 —figure supplement 2). In cells, such cleavage will affect the detection of higher oligomeric states much less due to the larger likelihood of detecting the remaining fluorescent proteins leading to observable colocalization. For monomers, however, this would lead to the exclusion of a larger proportion of proteins from the analysis. Because the recall rate of our fluorescent proteins is low, this would lead to the removal of predominantly non-colocalized clusters resulting in an overcounting of the remaining colocalized clusters. By including this probability as ‘*m*’, we were able to generate a reasonable fit to the observed distributions of the reference proteins.

Omitting ‘*m*’ is possible, especially for the determination of higher oligomeric states as seen in the revised Figure 4. However, this strongly increases the uncertainty of determining monomeric states and is thus not generally recommended (Figure 4 —figure supplement 1; Page 15).

4) The word stoichiometry in the title of the paper is misleading. Although the technique could be applied to measure the oligomerisation state of different subunits in independent samples using different expression constructs, and thus an average stoichiometry could be determined, it is not suitable to directly measure stoichiometries of different subunits in the same sample.

We agree and now use “oligomeric state” instead of stoichiometry in the title and throughout the manuscript. We changed it in the keyword list as well.

5) An open-source software tool would find wider-spread application and complement existing methods to measure the oligomerisation state of membrane proteins from monomers to tetramers using relatively standard PALM approaches.

We have implemented the most important methods of our manuscript (import from SNSMIL and SMAP, linear chromatic aberration correction, drift correction, extraction of the colocalization ratio, analysis of reference proteins to determine ‘*p*’ and ‘*m*’ as well as the analysis of proteins of interest using COM and Kolmogorov-Smirnov with bootstrap statistics) in a rather easy-to-use python library. Additionally, we provide example Jupyter notebooks that demonstrate the use of the library. All of this, including example data, can be found under https://www.github.com/GabStoelting/DCC-SMLM and is published under a GPL-3.0 license. We aim to maintain this library in the future and are happy to expand the capabilities further in collaboration with other users. The analyses performed in our manuscript, however, are still mostly based on older implementations of the algorithm and are still available under the link stated in the original manuscript (https://doi.org/10.5281/zenodo.6012450).

6) Some of the semantics need to be better defined for a more general readership.

We addressed this issue as stated below under the specific reviewer comments.

7) Regarding the fits to the data in Figure 4A, I assume that all of the (e.g., red) points were globally fit to the data for all 5 proteins with known stoichiometries using a single value of p. However, this is not stated in the text, which made it initially difficult for me to understand what the authors were doing here. Please describe the fitting in more detail in the text.

All points resulted from a global fit to the data. Based on the other comments, we now use global bootstrap resampling to obtain mean values and confidence intervals.

Reviewer #1 (Recommendations for the authors):1. I am under the impression that the approach requires identification of distinct proteins, hence the need for SMLM. The authors state that they achieve a 30nm radial resolution, so I assume that an inherent assumption here is that multiple proteins within 30nm of each other must be assumed to be rare? Even assuming this is ok, a 100nm cutoff is used for determining colocalization of spots in the two color channels. Why so much larger than 30nm? How do the authors ensure that only one protein is within each of these 100nm spots? Or can multiple PAmCherry spots be colocalized with the same mVenus spot, or vice-versa? This seems confusing to me. Or if this does not matter, please explain as the theory (e.g., Equation 3) requires counts of numbers of proteins. In the discussion, the authors suggest that their approach is actually not reliant on SMLM at all, and only requires "enough spatial resolution". Please define what "enough" means. And again, if resolving individual proteins is actually not required here, then this needs to be clarified.

Thank you for pointing this out. The 30 nm radial resolution (95% of localizations from a single cluster fall within a 30 nm radius around the cluster center) as shown in Figure 2B represents an optimal case. This resolution was determined using single color recordings of fluorescent beads following drift correction. However, the fluorescence from these beads is much brighter than the signals observed from fluorescent proteins increasing the precision of the localized coordinates. Furthermore, it does not consider the residual error following the correction of the lateral chromatic aberration in multi-color recordings. We determined radial distribution functions between the two fluorescent proteins tagged to the monomeric protein barttin and observed that the two (mVenus and PAmCherry) can be typically found within 100 nm of each other. We expanded our explanation to avoid ambiguities.

The spatial resolution of course also depends on the imaging modality. Although protein complexes must be more than 100 nm apart in DCC-SMLM experiments as implemented in our manuscript, more refined SMLM or STED microscopes may give higher resolution, allowing for complexes to be closer together. Other super-resolution techniques such as structured illumination or even conventional light microscopy (for example using *Xenopus oocytes*) might work if one can reduce the density of proteins so that individual protein complexes can be distinguished. There is no absolute requirement for spatial resolution built into the theory behind our DCC algorithm if colocalization can be observed and the resolution is high enough to provide spatial separation of protein complexes of interest. We clarify this aspect in the discussion.

2. Some of the semantics need to be better defined for a more general readership. What is a "localization"? Observation of a single fluorophore on a single frame, or the identified location of a single fluorophore across frames? Does a cluster of localizations represent a single protein, or a cluster of proteins? If it is the former, then requiring at least 3 localizations for a cluster to be analyzed (e.g., see Figure 5B) may limit background noise, but would also remove proteins where PAmCherry bleached within a couple of frames. What is the distribution of bleach times for PAmCherry, and what fraction are discarded by this cutoff? The authors suggest that this cutoff removes PAmCherry localizations that are likely to be background noise which do not colocalize with mVenus clusters. But if they do not colocalize, then how do they affect the computation at all, as I thought only colocalized clusters were considered? Overall, the methods should be described in more detail for a general readership.

We use “localization” to refer to the coordinate of an emission event localized by the SMLM software (SNSMIL in our case) on a single frame (of 85.59 ms duration) as well as the associated information (for example intensity and quality of fit). Within this manuscript, the term “cluster” is used for several localizations in close spatial proximity. We generally assume that they arise from emissions by fluorescent proteins located within the same diffraction-limited volume. Detailed definitions of these two terms have been implemented in the Results section of the manuscript.

We considered all clusters in the mVenus channel (where there are only very few background signals outside of cells), but only those PAmCherry clusters that are colocalized with mVenus clusters. As discussed under point (3) of the essential revisions, we did test for a range of minimum numbers of localizations per PAmChery cluster but going below 3 greatly reduced the apparent colocalization ratio. Such inappropriately small cutoff values for PAmCherry (Figure 5B) lead to the inclusion of a lot of background signals and this results in the exclusion of many true colocalizations by our background subtraction, lowering colocalization ratios (Refer to Equation 5, *N_MF_* is to be decreased to *N_MF_* * (1–*P_rb_*)). To ensure that only those sparsely distributed proteins are considered, our algorithm further excludes mVenus clusters that colocalize with more than one red cluster. On the other hand, if the cutoff value is only slightly smaller than the optimum and not lead to exclusion of many true colocalizations, the outcome should not be largely affected. However, if we increase the cutoff value for the red channel from its optimum, the contribution of both, background, and genuine signals, are expected to go down. Such reasoning is well supported by the analysis shown in Figure 5.

3. I have some reservations regarding the use of Equation 4, which to some extent appears to be a fudge factor given that the data does not quite fit the initial simple theory (Equation 1). First, it is my understanding that p encompasses all of the things that can lead to a fluorophore not being observed. So why then are some ways in which this could occur such as misfolding or truncation treated separately? Second, Equation 4 needs to be better explained as to why it is appropriate to describe misfolding or truncation events. Third, I would like to see Figure 4C-N repeated without using the fudge factor m. Is Equation 4 really needed to reliably determine the stoichiometries of the tested exchangers/transporters? And lastly, why should we expect m to be the same for different proteins? It seems to me that misfolding or truncation may be highly protein dependent. If m does differ from protein to protein, then it seems like this entire approach is no longer robust, at least for monomers.

We would like to thank Reviewer 1 for insisting on a better explanation for using the parameter ‘*m*’. Please see also our more detailed explanation under point (3) in the essential revision list above. In short, the factor ‘*p*’ only denotes the probability of PAmCherry being successfully detected but does not consider the possibility that some proteins have both, mVenus and PAmCherry fluorescence impaired. This would particularly affect monomeric proteins with only a single mVenus-PAmCherry tag. We repeated the analysis without ‘*m*’ and the results are similar but cannot longer give accurate predictions for monomeric proteins.

For all proteins, linkers between mVenus and C-terminus of the protein of interest or between PAmCherry and the C-terminus of mVenus were the same. Therefore, we do not anticipate a big variability of the factor *‘m’*, which describes the truncation and misfolding of both fluorescent proteins. In addition, our results do not suggest a high variability of ‘*m’* across our reference proteins. As stated above, we have revised our manuscript to better explain ‘m’ and discuss its limitations and requirements.

4. Equation 4 is introduced as a means to limit the contribution of background noise, but thereafter it appears that the authors just apply Equation 3 to their data. If so, what is the point of Equation 4, or is this a mistake in the text? Also, the variable N_CO in Equation 4 seems to be the same thing as N_MF in Equation 3? If so, please stick to one or the other, and if not please clarify.

We would like to thank the reviewer for this observation. Indeed, this a mistake in the text and we did use old Equation 4 (now Equation 5) rather than Equation 3 throughout the manuscript. *N_CO_* and *N_MF_* are indeed the same and we now use *N_MF_* in the new Equation 5 (old Equation 4).

5. Regarding the fits to the data in Figure 4A, I assume that all of the (e.g., red) points were globally fit to the data for all 5 proteins with known stoichiometries using a single value of p. However, this is not stated in the text, which made it initially difficult for me to understand what the authors were doing here. Please describe the fitting in more detail in the text.

As stated in the response to point (7) of the essential revision list, we did perform global fitting. We now added a bootstrap resampling procedure to determine mean values and confidence intervals and added a more detailed description.

6. The blue points in Figure 4A are hard to distinguish from the black points. Please choose a better representation. Also, for Figure 4A,B please show all of the data points rather than just box plots plus outliers. There are not so many data points to make this unreasonable in this case.

We adjusted Figure 4 accordingly.

Reviewer #2 (Recommendations for the authors):The detection, localisation and grouping of fluorescent events could be substantially improved by using maximum likelihood fitting of experimental point spread functions and post-filtering according to the log-likelihood ratio (LLR), as e.g. offered by the open source software SMAP (Ries, 2020, Nature Methods). This is expected to improve the detection of short fluorophore blinks while improving the rejection of background events.

As stated under point (2) of the essential revision list above, we now compare the result of SMAP with those of SNSMIL for some of our recordings. In short, we do not see a significant difference in the observed colocalization ratios, although we agree that more recent software (such as SMAP) should be used for future experiments.

The full potential, but also the limitations of the approach do not become clear because Equation 1 (or Equation 6) are not rigorously analysed with respect to their sensitivity/ability to differentiate different oligomerisation states. For example, it is clear that a high detection efficiency p renders the method insensitive. The optimal range p = 0.2-0.4 mentioned in the discussion is not substantiated. Also, the uncertainty of the experimentally determined values p and m could be accounted for by error propagation.

We would like to thank the reviewer for this comment. Please see our response to point

(1) of the essential revision list above for a detailed response. In short, we now provide a sensitivity analysis of Equation 6 as well as confidence intervals for all experimentally observed values, including error propagation required for the analysis of the “coefficient of mismatch” shown in Figure 4C-H. We have revised the text as well as including the sensitivity analysis as Figure 1 —figure supplement 1 and added the confidence intervals to Figure 4.

The manuscript would also benefit from more discussion regarding– The origin of the factor 'm'. Do HEK293T cells natively express any untagged protein corresponding to the transfected POIs? If not, it would be important to state this explicitly in the text. If yes, this violates the assumption of the analysis; in fact, it would be expected to contribute to the 'm' factor in equation 6 and lead to a significant variation of this factor from cell to cell, depending on the relative expression levels of tagged and untagged protein. Thus, a knock-out cell line needs first to be created before introducing the tagged POI.

Please see our response to point (3) on the essential revision list above for a detailed response regarding the origin of ‘*m*’. We can exclude a significant endogenous expression of our reference proteins in HEK293 cells, except for ClC-2 where we added another dimer, bovine ClC-K, to circumvent this problem. We have revised our manuscript to include a better explanation of ‘m’ and discussed the lack of endogenous expression of our reference proteins .

– The large variation observed from cell to cell. Figure 4, A and B, as well as Figure 4 —figure supplement 4 show substantial variation of the colocalization ratio P measured from cell to cell, as large as ranging from 0.15 to 0.45 (EAAT2) or 0.1 to 0.4 (CIC-2). The methods do not state how large the imaged/analysed FOV in a single cell was. According to Figure 1B, it was at least 5x5 µm^2^. With 1-2 clusters per µm^2^, this corresponds to 25-50 clusters. Together with the low detection efficiency p=0.12 for PAmCherry, this is expected to result in a substantial variability of P from measurement to measurement. This underlines how an improved detection efficiency, as could be achieved when using state-of-the-art SMLM, might translate in more accurate measurements.

We agree with this observation. Indeed, based on our current knowledge, we would suggest using slightly higher expression densities to record more clusters, thus reducing variability. We now include this thought in the discussion

.